# Assessing document section heterogeneity across multiple electronic health record systems for computational phenotyping: A case study of heart-failure phenotyping algorithm

**Sungrim Moon**[1], **Sijia Liu**[1], **Bhavani Singh Agnikula Kshatriya**[2], **Sunyang Fu**[1], **Ethan D. Moser**[3], **Suzette J. Bielinski**[3], **Jungwei Fan**[1], **Hongfang Liu**[1] *

**1** Department of Artificial Intelligence & Informatics, Mayo Clinic, Rochester, MN, United States of America,
**2** Department of Center for Digital Health, Mayo Clinic, Rochester, MN, United States of America,
**3** Department of Quantitative Health Sciences, Division of Epidemiology, Mayo Clinic, Rochester, MN, United States of America

* liu.hongfang@mayo.edu

## Abstract

### Background

The incorporation of information from clinical narratives is critical for computational phenotyping. The accurate interpretation of clinical terms highly depends on their associated context, especially the corresponding clinical section information. However, the heterogeneity across different Electronic Health Record (EHR) systems poses challenges in utilizing the section information.

### Objectives

Leveraging the eMERGE heart failure (HF) phenotyping algorithm, we assessed the heterogeneity quantitatively through the performance comparison of machine learning (ML) classifiers which map clinical sections containing HF-relevant terms across different EHR systems to standard sections in Health Level 7 (HL7) Clinical Document Architecture (CDA).

### Methods

We experimented with both random forest models with sentence-embedding features and bidirectional encoder representations from transformers models. We trained MLs using an automated labeled corpus from an EHR system that adopted HL7 CDA standard. We assessed the performance using a blind test set (n = 300) from the same EHR system and a gold standard (n = 900) manually annotated from three other EHR systems.

### Results

The F-measure of those ML models varied widely (0.00–0.91%), indicating MLs with one tuning parameter set were insufficient to capture sections across different EHR systems.

**Data Availability Statement:** Data sharing is restricted by the Mayo Clinic and Olmsted Medical Center Institutional Review Boards for human subject research because data from three different electronic health records contain sensitive patient information. Machine learning models will be available on request to rstnlp@mayo.edu that will provide data access request document.

**Funding:** This work was supported by National Institutes of Health grant numbers TR002062 (HL), LM011934 (HL), and HL136659 (SJB). This study was made possible using the resources of the Rochester Epidemiology Project, which was supported by the National Institute on Aging of the National Institutes of Health under Award Numbers AG034676 (Walter Rocca) and AG052425(Walter Rocca).

**Competing interests:** The authors have declared that no competing interests exist.

**Abbreviations:** BERT, Bidirectional Encoder Representations from Transformers; CDA, Clinical Document Architecture; EHR, electronic health records; GEC, General Electronic Centricity; HF, heart failure; HL7, Health Level 7; ML, machine learning; NLP, natural language processing; OMC, Olmsted Medical Center; REP, Rochester Epidemiology Project; RF, random forest.

The error analysis indicates that the section does not always comply with the corresponding standardized sections, leading to low performance.

## Conclusions

We presented the potential use of ML techniques to map the sections containing HF-relevant terms in multiple EHR systems to standard sections. However, the findings suggested that the quality and heterogeneity of section structure across different EHRs affect applications due to the poor adoption of documentation standards.

## Introduction

The wide adoption of electronic health records (EHRs) creates a rich and integrated data source for phenotypic information. Computational phenotyping, which automatically extracts phenotypes from EHR data, can accelerate the adoption and utilization of phenotype-driven efforts to advance scientific discovery and improve healthcare delivery. Given that much clinical information is embedded in clinical narratives, natural language processing (NLP) techniques have been extensively utilized to extract such information for accurate computational phenotyping. However, the interpretation of a term or phrase mentioned in a document depends on its associated section context. For instance, an occurrence of the term "heart failure" in the "Past Medical History" section most likely means the patient had a history of heart failure (HF). In contrast, "Brother had heart failure" in the "Family History" section would imply the patient has a family member with HF. Deploying computational phenotyping algorithms built upon one EHR system which incorporates section information to different EHR systems requires the mapping of relevant clinical sections due to the lack of standardization in clinical document practice [1, 2].

The eMERGE HF phenotyping algorithm was developed using data from the General Electronic Centricity (GEC) EHR system at Mayo Clinic, which adopted the Health Level 7 (HL7, http://www.hl7.org) Clinical Document Architecture (CDA) 1.0 standards [3]. The algorithm requires HF-relevant terms to be from clinical sections capturing patient information about current or past medical problems. Implementing the HF phenotyping algorithm in other EHR systems requires the accurate identification of those corresponding clinical sections. However, distinct EHR systems often use heterogeneous documentation (e.g., the lexical variants to describe the same clinical section, one major section that matches multiple granular document subsections in other EHR systems), leading to the challenge of identifying corresponding sections between separate EHRs for phenotype algorithms. In this study, we explored the use of embedding-based machine learning (ML) classifiers to detect corresponding sections among different EHRs, which were trained using a labeled corpus automatically extracted from the GEC system and evaluated using a blind test set sampled across four EHR systems using HF phenotyping algorithm as a case study. Two types of embedding-based ML classifiers, the random forest (RF) model and bidirectional encoder representations from the transformers (BERT) model, were experimented [4]. The performance evaluation of those classifiers also allowed us to assess the heterogeneity associated with the section information among them.

## Background

### Standardization of clinical section

Clinical documentation is quite complex as it can fall into different types (e.g., consultation notes, progress reports) depending on the purpose. For a given document type, sections or

subsections generally follow a logical sequence that has not changed much [5]. However, there are inconsistencies among different EHR vendors regarding document types and sections [6]. One effort to standardize clinical documentation is HL7 Clinical Document Architecture (CDA), initiated in 1996, one of the widely adopted HL7 version 3 standards. It standardizes document metadata and organizes clinical contexts into various sections. The latest Fast Healthcare Interoperability Resource (FHIR) specification (http://www.hl7.org/fhir) has adopted HL7 CDA as documentation standards. However, those standards have not been consistently adopted by EHR vendors [7, 8].

### Approaches for section detection

The detection of sections in a single EHR system or the alignment of corresponding sections and subsections across different EHR systems have been explored with diverse approaches such as rule-based, ML-based, or hybrid approaches [6]. For example, Melton et al. detected sections in operative notes using the regular expression with controlled Logical Observation Identifiers Names and Codes terminology (LOINC) [9]. Haug et al. utilized the HL7 CDA standard to ensure the level of CDA-compliance of their results after training the Bayesian network with N-gram features which were extracted from the section in pathology and radiology notes [10].

Recently, the state of the art NLP approaches such as embedding-based or deep learning techniques. Beyond overcoming the heterogeneity of diverse data types in EHR using a hierarchical embedding-based model, these approaches offer the potential to detect sections [6, 11–15]. For example, Sadoughi et al. used unidirectional long-short term memory (LSTM) units. And Salloum et al. proposed using bi-direction LSTM to detect sections while converting from medical dictations into clinical reports [16, 17]. In a recent study, Rosenthal et al. applied recurrent neural network (RNN) or the fine-tune BERT model using gated recurrence units trained with medical literature. They evaluated models using the Cleveland Clinic dataset and i2b2 dataset sentences to demonstrate the feasibility of detecting and classifying eleven common sections and sentences [18]. Their study showed that RNN or BERT could leverage the medical literature to predict clinical sections.

While section identification may enhance the performance of clinical NLP tasks, there is a lack of community efforts in standardizing clinical documentation structure.[6] As most clinical NLP studies are based on a single site, very few studies have investigated section detection across multiple EHR systems for computational phenotyping [6, 19, 20].

## Materials and methods

The overview of our methods for developing section identification classifiers and assessing the heterogeneity associated with the section information among multiple EHRs is shown in Fig 1.

### Resources

We used a training corpus consisting of clinical documents of 5,000 patients randomly selected from a primary care cohort. All clinical documents (n = 1.6 million) from 2009 to 2013 were retrieved from the Mayo Clinic GE Centricity (GEC) EHR. Among clinical documents, we randomly extract 100,000 clinical sections containing HF-relevant terms (i.e., "heart failure," "cardiac failure," "multi-organ failure," "ventricular failure," "CVF," and "LVF") [3]. As Mayo Clinic GEC EHR adopted HL7 CDA 1.0 standard, we had a weakly labeled silver standard for section identification [21].

We used a cardiovascular epidemiology cohort retrieved from the Rochester Epidemiology Project (REP) as the test set. REP is a record-linkage system capturing longitudinal patient

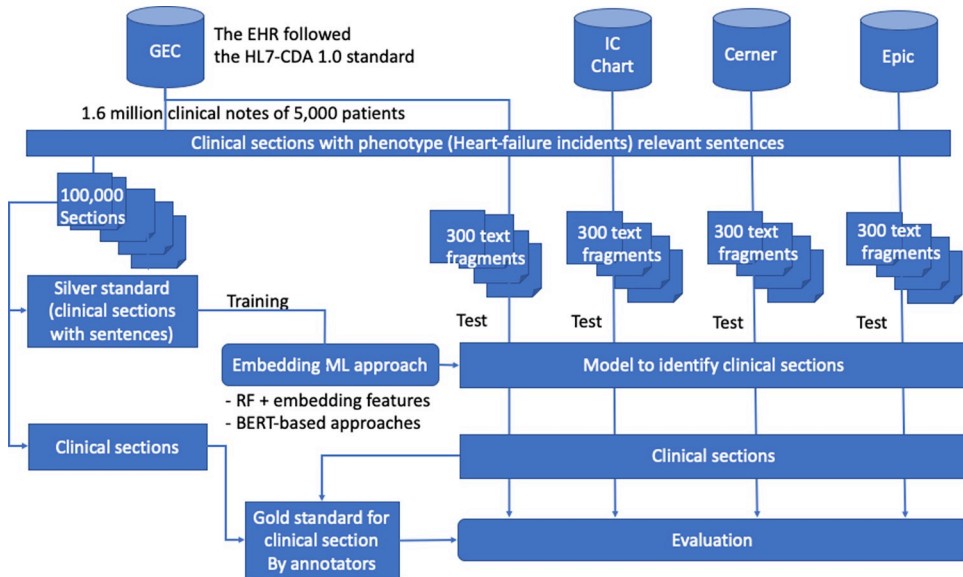

**Fig 1. The overview of assessing heterogeneity of clinical sections across three electronic health records using embedding-based machine learning approaches.** EHR = Electronic health records; HL7-CDA = Health Level 7—Clinical Document Architecture; ML = Machine Learning; RF = Random Forest; BERT = Bidirectional Encoder Representations from Transformers; GEC = General Electronic Centricity EHR.

records of a cohort of 750,000 patients from various health care institutions, drawn from the population who resided in 27 counties in southern Minnesota and western Wisconsin between 1998 and 2019 [22–24].

The study has appropriate approvals by the Mayo Clinic and Olmsted Medical Center Institutional Review Boards (IRBs Mayo 13–009317, Mayo 17–008818, and OMC 053-OMC-17).

## Gold standard creation

To create the gold standard annotation, we retrieved 900 text segments from three EHR systems (300 each from IC Chart, Cerner, and Epic EHR systems) at two medical centers, Mayo Clinic and Olmsted Medical Center. The goal was to assess the performance of section identification classifiers and the heterogeneity across different EHR vendors. Those 900 text segments were randomly drawn from all text segments containing HF-relevant terms [3]. S1 Table presents examples of sentences with potential sections containing the HF-relevant terms from individual EHR systems, where the potential sections were retrieved through heuristic rules. As the eMERGE HF algorithm was developed based on the GEC EHR incorporating HF-relevant terms from six specific sections (i.e., "Assessment and Plan," "History of Present Illness," "Past Medical History," "Chief Complaint and Reason for Visit," "Problem," and "Review of systems" sections), it is crucial to identify those relevant sections in those other EHR systems in order to implement HF phenotyping algorithm [3].

Two trained abstractors (Ethan D Moser and Donna Ihrke) annotated the 900 text segments by assigning a section or subsubsection to each occurrence of the HF-relevant terms and mapping them to standard sections of the HL7 CDA 1.0 (i.e., sections used in the GEC EHR). The section mapping can be fully or partially. If a given text segment can be mapped to an HF phenotyping-relevant section in the GEC EHR, the mapping is considered fully. Otherwise, it is considered partially mapping with all possible sections listed according to the high relevance of the context. For example, one text segment from the "subjective" sections in the

> For Segment *Seg* consisting of *s* sentences {$S_1$, $S_2$, $S_3$, …, $S_s$}
>
> Six binary section classifiers:
>
> For each Section *C* from {$C_1$, $C_2$, …, $C_6$}
>
> Classify ($S_i$, *C*) as true or false
>
> Aggregate (*Seg*, *C*) as true or false across *s* sentences. If any of them returns true, the output
>
> is true. Otherwise, the output is false. In case of a tie, randomly assign true or false.
>
> Heart-failure phenotyping-relevant section classifier:
>
> If any of the six binary section classifiers return true, output true. Otherwise, output false.

**Fig 2. The algorithm of section classifiers.**

other EHR system is mapped partially to the "History of Present Illness" and "Chief Complaint and Reason for Visit" sections in the GEC EHR. The third independent nurse abstractor (Ellen E. Koepsell) adjudicated the discrepant cases between the other two abstractors. If there were multiple mappings, fully or partially, we choose the most relevant section for the given text segment. The inter-annotator reliability was assessed using percent agreement and Cohen's kappa.

## Algorithm for the section classification

A total of seven classifiers were developed (as shown in Fig 2). For example, in the following text segment from the problem section, "DIAGNOSIS: 1. Congestive Heart Failure. 2. Chronic Systolic (Congestive) Heart. 3. Anemia.", *Seg* consists of three sentences, {*"1. Congestive Heart Failure," "2. Chronic Systolic (Congestive) Heart," "3. Anemia"*}. The six binary classifiers classify the text segment and the specific section as true or false for each section *C* (*"Assessment and Plan," "History of Present Illness," "Past Medical History," "Chief Complaint and Reason for Visit," "Problem," or "Review of systems"*). The above example with the "Problem" classifier predicts whether each of the three sentences is the "Problem" section or not. If the classifier labels any of the three sentences belonging to the "*Problem*" section, it results in output as true, and then the text segment is labeled as the "*Problem*" section, which is a case of true positive for the "Problem" classifier. Meanwhile, if the "History of Present Illness" classifier labels any of the three sentences as true, then the text segment is labeled as the "*History of Present Illness*" section, which is a case of false positive for the "History of Present Illness" classifier because the original sentences in the silver standard belong to the problem section. The HF phenotyping-relevant section classifier combined the outputs of six ML classifiers to make a decision collectively. If any of the six classifiers output true, i.e., *Seg* belongs to one of the HF phenotyping-relevant sections, the classifier output is true. In the above example, the output of the HF phenotyping-relevant section classifier is true because the "Problem" classifier output is true.

Two types of embedding-based ML algorithms were compared, a random forest (RF) model with an embedded-encoding sentence using Bert-as-service (https://github.com/hanxiao/bert-as-service) and a pre-trained clinical BERT model (https://github.com/google-research/bert). Bert-as-service used BERT-base-uncase pre-trained model as a sentence encoder with the "REDUCE_MEAN" strategy by converting each input sequence (a sentence using the NLTK sentence tokenizer) into a 768-dimensional vector [25]. For random forest model, we used the default threshold as 0.5. Our BERT model started from the clinical BERT-

base-uncased pre-trained model and fine-tuned for four epochs, with hyper-parameter settings of 32 batch size, 3e-5 learning rate, and 512 max sequence length. All training sentences for both models were in the silver standard; the clinical notes from Mayo Clinic GEC EHR adopted HL7 CDA 1.0 standard. Note that we choose binary classifiers due to the low performance of a multi-class or multi-label classifier.

## Evaluation

We evaluated the performance of the section identification classifiers using a test set of 300 text segments from the GEC and a manually annotated gold standard as described above. The precision, recall, and F-measure were computed. We focused on assessing the interesting individual section rather than macro or weighted average evaluation because we used random phrases corresponding to HF-relevant sections in the test EHRs. If we denote true positives, true negatives, false positives, and false negatives as *TP*, *TN*, *FP*, *and FN*, respectively, those metrics are defined as follows:

$$Precison = \frac{TP}{TP + FP}, \ Recall = \frac{TP}{TP + FN}, \ F - measure = 2 * \frac{precision * recall}{precision + recall} \qquad (1)$$

## Results

The training set (from GEC EHR) consists of 47 unique sections from randomly selected 100,000 sections containing HF terms. The clinical section distribution in the training set is 39% for "Assessment and Plan," 23% for "History of Present Illness," 11% for "Problem," 7% for "Past Medical History," 3% for "Chief Complaint and Reason for Visit," less than 1% (0.89%) of "Review of systems," and 16% of other 41 sections. The test set consists of a total of 110 unique sections and subsections from 900 text segments from three different EHRs (IC Chart, Cerner, and Epic). The percentage agreement and Cohen's kappa between the two annotators was 78% and 0.56 (agreed 79 corresponding sections/subsections among 110 clinical sections) for those fully mapped, versus 80% and 0.35 (agreed 100 corresponding sections/subsections among 110 clinical sections) for those partially mapped.

Based on the manual annotations, the section headers of the three other EHR systems show a high degree of variety in representing patient-specific HF-relevant sections. For example, we identified the GEC section, "Assessment and Plan" had 23 different expressions from the other three EHR systems (22 fully corresponding sections and one partially corresponding section) in Table 1. The most frequent partially corresponding section is the "History of Present Illness." (e.g., the "Disease Summary" subsections partially corresponded to the "History of Present Illness"). During the generation of the gold standard, a given text segment of the "Problem" section in the other three EHRs is often partially mappable to the following sections in the GEC EHR, "Chief Complaint and Reason for Visit," "Past Medical History," and "Others (Consults)" depends on the context as well as the viewpoint of the annotator.

The distribution of clinical sections in the test sets is shown in Fig 3. Table 2 shows the general statistics of the test sets. We observed HF-relevant terms appear more frequently in the "Assessment and Plan," "History of Present Illness," and "Problem." The Cerner corpus has nearly 46% (gray bar in Fig 3) samples containing HF-relevant terms in the "Problem" section. In contrast, the GEC and Epic (blue and yellow bars in Fig 3) have the majority in "Assessment and Plan" and "History of Present Illness." In the case of the IC Chart (orange in Fig 3), "Assessment and Plan" is the majority, followed by "Past Medical History" and "History of Present Illness." Overall, the GEC corpus contained most HF-relevant terms in those six specific sections (96% in Table 2), while other sets had relatively high HF-relevant terms in the "Other" sections (ranging from 7% to 11%).

**Table 1. Corresponding sections and subsections among test corpora.**

| Sections of GEC EHR | Sections and subsections in IC Chart, Cerner, and Epic EHRs (the gold standard) | | | | |
|---|---|---|---|---|---|
| | **Fully corresponding sections/subsections** | | **Partially corresponding sections/subsections** | | |
| **Sections** | **N** | **Section and subsections** | **N** | **Section and subsections** |
| **Assessment and Plan** | 22 | 'A', 'A/P', 'Problem List Items addressed this Visit', 'Plans', etc. | 1 | 'Radiology Tests to be Ordered' |
| **History of Present Illness** | 11 | 'Active Medical Issues', 'History', 'S', 'Summary', etc. | 5 | 'Background', 'CC / Subjective', 'Clinical Synopsis', 'Disease Summary', etc. |
| **Past Medical History** | 10 | 'Past Medical/Family/Social/Surgical History', 'Brief Medical History', etc. | 3 | 'Diagnosis History', 'Medical History', 'Medical Illnesses' |
| **Chief Complaint and Reason for Visit** | 5 | 'Chief Complaint', 'Reason for Admission', 'Reason for Visit, etc. | 3 | 'Problem', 'Problems', 'Visit Note' |
| **Problem** | 7 | 'Diagnoses and all orders for this visit', 'Diagnosis Plan', 'Visit Diagnoses', etc. | 3 | 'Admission Diagnosis', 'Diagnostic Studies', 'Referral Diagnosis' |
| **Review of systems** | 3 | 'Cardiovascular', 'Review of Systems' 'ROS' | 3 | 'Cadriovascular Risk' (typo in subsection), 'Cardiovascular Risk', 'Objective' |
| **Others[a]** | 21 | 'Current Medications', 'Education Discussion', 'Referrals/Consultations', etc. | 13 | 'Contact Anesthesiologist Regarding', 'Heart', 'PFSH', 'Procedure', 'Recent Investigations', etc. |

GEC = General Electronic Centricity; EHR = Electronic Health Records.

[a]Non-heart failure relevant sections for heart failure algorithm. *N*: numbers of section and subsections.

The section detection performance of the embedding-based MLs is shown in Table 3. It varies widely, from 0.00 to 0.91 F-measure in Table 3. The performance within the homogenous EHR (i.e., the test set from GEC EHR) is higher than those of heterogeneous EHR (i.e., the test sets from the other three EHRs) except in the "Past Medical History" section for BERT models. Both RF and BERT models show higher performance in identifying "HF phenotyping-relevant sections" (i.e., six specific sections) than individual sections. Especially, the BERT models can capture "HF phenotyping-relevant sections" sections with the best F-measure as 0.88 for the GEC corpus and 0.81–0.86 for three other EHRs. Also, the BERT models tend to achieve better performance than the RF models with embedding-based features. However, both RF and

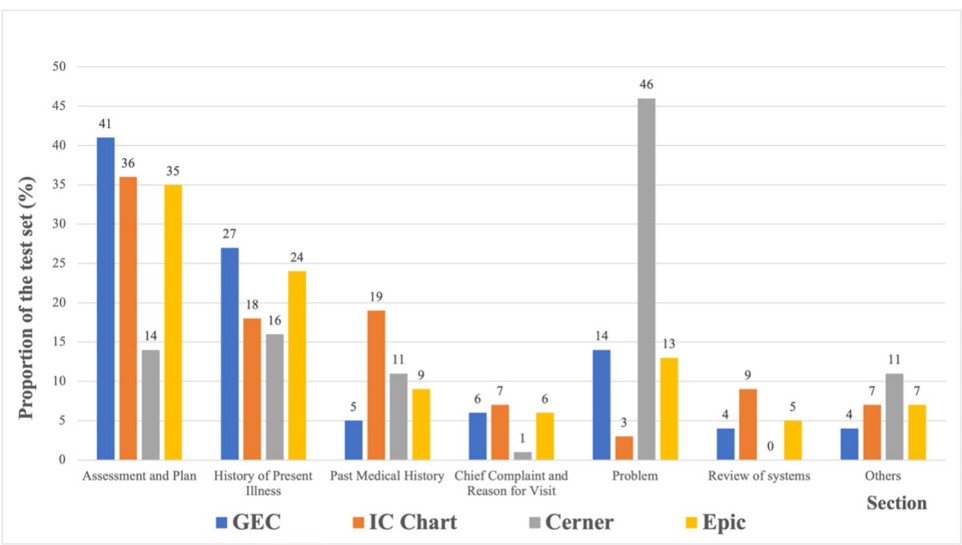

**Fig 3. The distribution of the sections and subsections in the test.** GEC = General Electronic Centricity EHR.

**Table 2. Case[a] and non-case[b] for train set in GEC and the heart failure relevant clinical section in test sets.**

| Section | Train set in GEC (n = 90000) | | Test set in GEC (n = 300) | | Test set in three different corpora | | | | | |
|---|---|---|---|---|---|---|---|---|---|---|
| | | | | | IC Chart (n = 300) | | Cerner (n = 300) | | Epic (n = 300) | |
| | Case (%) | Non-case (%) | Case (%) | Non-case (%) | Case (%) | Non-case (%) | Case (%) | Non-case (%) | Case (%) | Non-case (%) |
| **Assessment and Plan** | 35244 (39%) | 54756 (61%) | 122 (41%) | 178 (59%) | 108 (36%) | 192 (64%) | 41 (14%) | 259 (86%) | 106 (35%) | 194 (65%) |
| **History of Present Illness** | 20950 (23%) | 69050 (77%) | 81 (27%) | 219 (73%) | 55 (18%) | 245 (82%) | 49 (16%) | 251 (84%) | 73 (24%) | 227 (76%) |
| **Past Medical History** | 6519 (7%) | 83481 (93%) | 15 (5%) | 285 (95%) | 57 (19%) | 243 (81%) | 33 (11%) | 267 (89%) | 27 (9%) | 273 (91%) |
| **Chief Complaint and Reason for Visit** | 2355 (3%) | 87645 (97%) | 17 (6%) | 283 (94%) | 20 (7%) | 280 (93%) | 4 (1%) | 296 (99%) | 19 (6%) | 281 (94%) |
| **Problem** | 10030 (11%) | 79970 (89%) | 259 (86%) | 259 (86%) | 10 (3%) | 290 (97%) | 138 (46%) | 162 (54%) | 40 (13%) | 260 (87%) |
| **Review of systems** | 775 (1%) | 89225 (99%) | 288 (96%) | 288 (96%) | 28 (9%) | 272 (91%) | 1 (0%) | 299 (100%) | 15 (5%) | 285 (95%) |
| **HF phenotyping-relevant** | 75873 (84%) | 14127 (16%) | 12 (4%) | 12 (4%) | 278 (93%) | 22 (7%) | 266 (89%) | 34 (11%) | 280 (93%) | 20 (7%) |

GEC = General Electronic Centricity; HF = Heart Failure.

[a] a sentence chunk containing the description of a heart failure incident

[b] a sentence chunk not about a heart failure incident

BERT failed to capture the "Problem" section. The "Chief Complaint and Reason for Visit" and "Review of system" sections were also not captured by RF. Note that the performance of the RF and BERT models for the "Review of systems" section in the Cerner EHR is 1.00 for precision (Table 3); however, it has less meaning because the total sample size of the case is one in Table 2.

## Discussion

The generalizability of NLP-empowered computational phenotyping algorithms leveraging section information depends on how well those sections are aligned across different EHR systems. We developed and evaluated embedding-based approaches (e.g., derived embedded features or utilizing available BERT models) for mapping clinical sections containing HF-relevant terms across four EHR systems. We also investigated the transferability of section classifiers trained with a labeled corpus to different EHR systems with the following key findings: 1) models for section classification trained using a single EHR system have limited generalizability to other EHR systems because of different section structures. For computational phenotyping algorithms leveraging section information (e.g., the eMERGE HF phenotyping algorithm), the heterogeneity associated with section structure may require dedicated efforts to account for such heterogeneity when deploying NLP-based phenotyping algorithms across different sites. 2) significant variation in clinical documentation across different EHR systems limits the full potential of using EHR for clinical research as each site may need to develop site-specific computational phenotyping algorithms. According to our knowledge, our study is the first study that quantitatively assessed the impact of documentation heterogeneity across multiple EHR systems for computational phenotyping.

In general, we observed a certain level of transferability across different EHR systems, especially for the "Assessment and Plan" section and the "History of Present Illness" section. It is known that the most informative resources for medical experts to obtain the comprehensive medical history of patients were in the "Assessment and Plan" section as well as the "History of

**Table 3. Evaluation of binary embedding-based ML models regarding the section.**

| ML | Section | Test set in GEC | Test set in three different EHRs | | |
|---|---|---|---|---|---|
| | | | IC Chart | Cerner | Epic |
| | | P (R) | P (R) | P (R) | P (R) |
| | | F (95% CI) | F (95% CI) | F (95% CI) | F (95% CI) |
| RF | Assessment and Plan | 0.53 (0.98) | 0.41 (0.88) | 0.15 (0.85) | 0.38 (0.90) |
| | | 0.69 (0.64,0.74) | 0.56 (0.50,0.62) | 0.26 (0.21,0.31) | 0.54 (0.48,0.59) |
| | History of Present Illness | 0.49 (0.88) | 0.42 (0.87) | 0.48 (0.92) | 0.43 (0.74) |
| | | 0.63 (0.57,0.68) | 0.56 (0.51,0.62) | 0.63 (0.57,0.68) | 0.55 (0.49,0.60) |
| | Past Medical History | 0.38 (0.33) | 0.26 (0.35) | 0.19 (0.12) | 0.19 (0.48) |
| | | 0.36 (0.30,0.41) | 0.30 (0.24,0.35) | 0.15 (0.11,0.19) | 0.28 (0.23,0.33) |
| | Chief Complaint and Reason for Visit | 1.00 (0.12) | 0.11 (0.05) | 0.00 (0.00) | 0.00 (0.00) |
| | | 0.21 (0.16,0.26) | 0.07 (0.04,0.10) | 0.00 | 0.00 |
| | Problem | 0.60 (0.07) | 0.00 (0.00) | 0.00 (0.00) | 0.00 (0.00) |
| | | 0.13 (0.09,0.17) | 0.00 | 0.00 | 0.00 |
| | Review of systems | 1.00 (0.42) | 1.00 (0.07) | 0.00 (0.00) | 0.50 (0.07) |
| | | 0.59 (0.53,0.64) | 0.13 (0.09,0.17) | 0.00 | 0.12 (0.08,0.15) |
| | HF phenotyping-relevant | 0.93 (0.83) | 0.88 (0.94) | 0.79 (0.87) | 0.89 (0.94) |
| | | 0.88 (0.84,0.91) | 0.91 (0.87,0.94) | 0.83 (0.79,0.87) | 0.91 (0.88,0.94) |
| BERT | Assessment and Plan | 0.94 (0.84) | 0.83 (0.54) | 0.41 (0.73) | 0.72 (0.62) |
| | | 0.89 (0.86,0.93) | 0.65 (0.58,0.72) | 0.52 (0.47,0.58) | 0.67 (0.61,0.72) |
| | History of Present Illness | 0.89 (0.81) | 0.57 (0.53) | 0.69 (0.92) | 0.69 (0.59) |
| | | 0.85 (0.81,0.89) | 0.55 (0.49,0.60) | 0.79 (0.74,0.84) | 0.64 (0.58,0.69) |
| | Past Medical History | 0.47 (0.47) | 0.61 (0.39) | 0.45 (1.00) | 0.56 (0.81) |
| | | 0.47 (0.41,0.52) | 0.47 (0.42,0.53) | 0.62 (0.56,0.67) | 0.67 (0.61,0.72) |
| | Chief Complaint and Reason for Visit | 0.75 (0.71) | 0.20 (0.50) | 0.06 (0.50) | 0.38 (0.32) |
| | | 0.73 (0.68,0.78) | 0.28 (0.23,0.33) | 0.11 (0.07,0.14) | 0.34 (0.29,0.40) |
| | Problem | 0.91 (0.24) | 0.00 (0.00) | 1.00 (0.01) | 0.00 (0.00) |
| | | 0.38 (0.33,0.44) | 0.00 | 0.03 (0.01,0.05) | 0.00 |
| | Review of systems | 1.00 (0.58) | 0.60 (0.32) | 0.25 (1.00) | 1.00 (0.33) |
| | | 0.74 (0.69,0.79) | 0.42 (0.36,0.47) | 0.40 (0.34,0.46) | 0.50 (0.44,0.56) |
| | HF phenotyping-relevant | 1.00 (0.79) | 0.93 (0.72) | 0.91 (0.82) | 0.95 (0.71) |
| | | 0.88 (0.85,0.92) | 0.81 (0.77,0.85) | 0.86 (0.82,0.90) | 0.81 (0.77,0.86) |

ML = Machine Learning; GEC = General Electronic Centricity; EHR = Electronic Health Records; RF = Random Forest; BERT = Bidirectional Encoder Representations from Transformers; HF = Heart Failure; P = Precision; R = Recall; F = F-measure.

Present Illness" section [26]. Therefore, section classifiers for those two sections have better performances due to the necessity of having higher consistency in documenting critical information. Meanwhile, the "Review of System" does not sufficiently record medical history [26]. The distribution of HF-relevant concept mentions reflects similar observations as those of the previous study [26].

Our study demonstrates the mapping of clinical sections across different EHRs is a challenging task, even though the mapping of specific clinical sections to standards does not impact much on the eMERGE HF algorithm. Specifically, the classifier for detecting HF phenotyping-relevant sections achieved high performance as HF-relevant terms tend to occur only in certain sections in those EHR systems. In our training and test sets, about 16% and 7% of HF-relevant terms appeared in clinical sections that were not HF phenotyping-relevant sections, respectively. In this study, we achieved an above 0.8 F1-score for detecting HF

phenotyping-relevant sections that present high transferability. However, these may not be true for other phenotypes (e.g., pain).

Both BERT and RF models failed for the "Problem" section. GEC's "Problem" section consists of noun phrases, incomplete sentences, or short expressions rather than a complete sentence structure. The average number of words in a sentence in the training set (i.e., the GEC EHR) for the "Problem" section is less than 25. Those embedding-based techniques might fail as little contextual information was available to learn. This inferior result was analogous to preliminary results of prior studies, subsequently resolved through the wide-ranging parameter tuning and class optimization [27]. Meanwhile, the "Problem" section in other EHR systems contains full sentence structures. According to our error analysis, the majority of sentences in the "Problem" section were incorrectly categorized into the "Chief Complaint and Reason for Visit" or "Past Medical History" sections, which often contained repeated information about the disease information of patients in the GEC EHR. Additionally, our training data for the "Problem" section is very skewed, with only 4.76% of sentences labeled as positive (i.e., from the "Problem" section).

We observed the lack of section standardization in clinical documentation might contribute to the low inter-annotator agreements (i.e., fair and moderate agreements). For example, family history information can be documented in the "Chief Complaint and Reason for Visit" section. Additionally, we noticed many subsections were defined but not mappable to standards, similar to prior studies [10]. This implies significant challenges faced in detecting section context for clinical NLP. As the prior study presented, the HL7 CDA standards provide diverse representations to capture certain granular information within the various sentences [28]. If clinical sections could conform more consistently to standards like the HL7 CDA, it will significantly raise the value of EHR in the secondary use for research.

Our model demonstrated the wide range of transferability of section information regarding patients' HF information associated with the medical history across four different EHR systems. The distribution of HF phenotyping relevant sections varies across EHRs. For example, the most frequent section for the IC chart and Epic is the "Assessment and Plan" section, with 36% and 35% prevalence, respectively. In comparison, the most frequent section for the Cerner EHR is the "Problem" section (46%). Additionally, the "History of Present Illness" section of the Epic EHR contains more HF-relevant terms (24%) compared to the other two EHR systems (18% for the IC chart and 16% for Cerner). HF-relevant terms were prevalent in the "Assessment and Plan" and "History of Present Illness" sections in the GEC EHR but with low occurrences in other EHRs, as shown in Table 2 and Fig 3. The section distribution variation across different EHR systems for HF-relevant terms may also be due to the characteristics of the practice settings rather than the EHR systems alone.

Note that our study focused on the assessment of heterogeneity of section information across different EHRs through the performance evaluation of embedding-based section classifiers. We trained ML models for section classification using a large corpus consisting of CDA-compliant clinical documents from one EHR system and tested on text fragments containing HF-relevant terms from multiple other EHR systems. We did not thoroughly compare various ML algorithms and other BERT-based approaches for section classification tasks or exhausting parameter optimizations for the RF and BERT models. Furthermore, the sample size of the test set (n = 1,200) was limited, which may lead to high variations. To fair comparison, we used the same chunk of the sentence set for training both BERT and RF models; however, this setting may lead to the disadvantage of BERT, which is a contextualized language model. The study confirmed poor adoption of standardization in clinical documentation, which can cause significant challenges for computational phenotyping leveraging section information. Additionally, we only conducted the assessment on one computational phenotyping algorithm, but the

evidence of the heterogeneity is clear. Our future direction would be to explore context-aware computational phenotyping algorithms to infer the section information from the local discourse context rather than leveraging section headers.

## Supporting information

**S1 Table. Differentiation of corresponding sections and subsections among corpora.** [a]One clinical document standard of IC Chart Electronic Health Record (EHR) consists of sections 'S,' 'O', 'A,' and 'P.'; The "Impression and Plan" section of the Cerner EHR, corresponds to "the Assessment and Plan" section of the General Electronic Centricity (GEC) EHR, which contains information on the "Problem," "Medications," and other sections; "Chief Complaint and Reason for Visit" sections of the Epic EHR are similar to "Problem" sections of the GEC EHR.
(DOCX)

## Acknowledgments

We would like to thank Donna Ihrke and Ellen E. Koepsell for conducting annotations, as well as Dr. Walter Rocca for providing the dataset.

## Author Contributions

**Data curation:** Sungrim Moon, Ethan D. Moser.

**Formal analysis:** Sungrim Moon.

**Investigation:** Suzette J. Bielinski, Hongfang Liu.

**Methodology:** Sungrim Moon.

**Software:** Sungrim Moon, Sijia Liu, Bhavani Singh Agnikula Kshatriya, Sunyang Fu.

**Supervision:** Hongfang Liu.

**Writing – original draft:** Sungrim Moon.

**Writing – review & editing:** Sungrim Moon, Bhavani Singh Agnikula Kshatriya, Suzette J. Bielinski, Jungwei Fan, Hongfang Liu.

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
