## [Decision Letter · Decision Letter 0]

15 Nov 2022

PONE-D-22-24883Assessing document section heterogeneity across multiple electronic health record systems for computational phenotypingPLOS ONE

Dear Dr. Moon,

Thank you for submitting your manuscript to PLOS ONE. After careful consideration, we feel that it has merit but does not fully meet PLOS ONE’s publication criteria as it currently stands. Therefore, we invite you to submit a revised version of the manuscript that addresses the points raised during the review process.

We look forward to receiving your revised manuscript.

Kind regards,

William Speier

Academic Editor

PLOS ONE

Journal Requirements:

 "This work was supported by National Institutes of Health grant numbers TR002062 (HL), LM011934 (HL), and HL136659 (SJB). This study was made possible using the resources of the Rochester Epidemiology Project, which was supported by the National Institute on Aging of the National Institutes of Health under Award Numbers AG034676 (Walter Rocca) and AG052425(Walter Rocca)."

Additional Editor Comments:

Both reviewers noted the value of the study described in this manuscript. However, they both also mentioned some areas for clarification an improvement. In particular, additional description of the methodology and experimental design as well as some additional discussion of previous work in this area could be beneficial.

Reviewers' comments:

Reviewer's Responses to Questions

**Comments to the Author**

1. Is the manuscript technically sound, and do the data support the conclusions?

Reviewer #1: Partly

Reviewer #2: Partly

2. Has the statistical analysis been performed appropriately and rigorously? 

Reviewer #1: No

Reviewer #2: No

3. Have the authors made all data underlying the findings in their manuscript fully available?

Reviewer #1: No

Reviewer #2: No

4. Is the manuscript presented in an intelligible fashion and written in standard English?

Reviewer #1: Yes

Reviewer #2: Yes

5. Review Comments to the Author

Reviewer #1: 1. You may need to cite or compare this work as it utilized advanced machine learning models to leverage heterogeneous EHR "HCET: Hierarchical Clinical Embedding With Topic Modeling on Electronic Health Records for Predicting Future Depression"

2. How did choose the threshold for the F-measure? It is more robust to show average precision or precision-recall area under the curve instead of F-measure as the latter depends on the threshold you choose

3. What's the ratio between case and non-case in for the training set of GEC? Because the ratio of cases is above 90%in all test sets, which is quite high and not reflect the true ratio in reality for the classifier. You need to resample the test set

4. Since both algorithms generated high F1 score (above 0.8) in the final binary outcome of HF phenotyping-relevant, you can draw the conclusion of transferability instead of focusing on each subsection.

5. In line 79, please remove "embedding-based" as random forest can take any input types. Writing in this way would make readers misunderstand that it inherently takes embeddings as the input.

6. In line 177, it should be true positive instead of false positive.

Reviewer #2: In this paper the authors evaluated the use of machine learning algorithms to detect heterogeneity for heart failure phenotyping across different EHR systems. Specifically, the authors used RF and BERT to classify sections given a segment of multiple sentences from an EHR note. The authors defined a transferability metric and showed that the models trained on one EHR system performed differently on other EHR-systems and the transferability score also varied across different EHR systems.

Strengths:

=======

1. The paper is well-written and well-structured.

2. The authors demonstrated the proper use of multiple EHR databases.

3. The authors showed the complexity and heterogeneity of heart-failure relevant section names across different EHR systems.

Weaknesses:

=======

1. The algorithm for the section classification is not clear. Why the authors preferred multiple separate classifiers instead of a single multi-class or multi-label (if a text segment belongs to multiple sections) classifier? Why all sentences in a text segment were classified separately (this weakens contextualized language models such as BERT)? How was the model training performed for these separate classifiers?

2. The transferability metric is not well-defined. A model with similar F-scores on two different test sets does not necessarily mean the system is transferable. A better evaluation method is to train a model (say, BERT) on a training set of the “to-be-tested” EHR system (IC Chart, Cerner, or Epic), test it on its hold-out test set and then compare this performance with the performance of a similar model trained on the source EHR system (GEC EHR). This gives a good indication of transferability, also known as domain adaptation.

3. The authors should explain why difference in section names among different EHR systems is analogous to overall heterogeneity. The paper title suggests overall heterogeneity and do not indicate that the study is limited to heart failure phenotyping algorithm. Either these should be explained in the paper body or the title should be revised.

4. The term ‘case’ was not defined for table-2.

6. PLOS authors have the option to publish the peer review history of their article (what does this mean?). If published, this will include your full peer review and any attached files.

Reviewer #1: No

Reviewer #2: No

---

## [Author Response · Author response to Decision Letter 0]

16 Feb 2023

We would like to thank the editor and the reviewers for their valuable feedback on our manuscript. Here, we revised the manuscript to address all comments. For detail, please see the response to the reviewers' file (ResponseToReviewers.docx)

---

## [Decision Letter · Decision Letter 1]

20 Mar 2023

Assessing document section heterogeneity across multiple electronic health record systems for computational phenotyping: A case study of heart-failure phenotyping algorithm

PONE-D-22-24883R1

Dear Dr. Moon,

We’re pleased to inform you that your manuscript has been judged scientifically suitable for publication and will be formally accepted for publication once it meets all outstanding technical requirements.

Kind regards,

William Speier

Academic Editor

PLOS ONE

Additional Editor Comments (optional):

Reviewers' comments:

Reviewer's Responses to Questions

**Comments to the Author**

1. If the authors have adequately addressed your comments raised in a previous round of review and you feel that this manuscript is now acceptable for publication, you may indicate that here to bypass the “Comments to the Author” section, enter your conflict of interest statement in the “Confidential to Editor” section, and submit your "Accept" recommendation.

Reviewer #1: All comments have been addressed

Reviewer #2: All comments have been addressed

2. Is the manuscript technically sound, and do the data support the conclusions?

Reviewer #1: Yes

Reviewer #2: Yes

3. Has the statistical analysis been performed appropriately and rigorously? 

Reviewer #1: Yes

Reviewer #2: Yes

4. Have the authors made all data underlying the findings in their manuscript fully available?

Reviewer #1: No

Reviewer #2: No

5. Is the manuscript presented in an intelligible fashion and written in standard English?

Reviewer #1: Yes

Reviewer #2: Yes

6. Review Comments to the Author

Reviewer #1: (No Response)

Reviewer #2: (No Response)

7. PLOS authors have the option to publish the peer review history of their article (what does this mean?). If published, this will include your full peer review and any attached files.

Reviewer #1: No

Reviewer #2: No

---

## [Editor Report · Acceptance letter]

23 Mar 2023

PONE-D-22-24883R1 

Assessing document section heterogeneity across multiple electronic health record systems for computational phenotyping: A case study of heart-failure phenotyping algorithm 

Dear Dr. Moon:

I'm pleased to inform you that your manuscript has been deemed suitable for publication in PLOS ONE. Congratulations! Your manuscript is now with our production department. 

Kind regards, 

on behalf of

William Speier 

Academic Editor

PLOS ONE